# The Effect of Transoesophageal Echocardiography on Treatment Change in a High-Volume Stroke Unit

**DOI:** 10.3390/jcm10040805

**Published:** 2021-02-17

**Authors:** Camille Siegel, Benjamin Marchandot, Kensuke Matsushita, Antonin Trimaille, Corina Mirea, Marilou Peillex, François Sauer, Cecile How-Choong, Floriane Zeyons, Olivier Rouyer, Annie Trinh, Helene Petit-Eisenmann, Laurence Jesel, Patrick Ohlmann, Valérie Wolff, Olivier Morel

**Affiliations:** 1Division of Cardiovascular Medicine, Hôpital Civil, Strasbourg University Hospital, 67000 Strasbourg, France; c-siegel@hotmail.fr (C.S.); benjaminmarchandot@gmail.com (B.M.); matsuken_22@yahoo.co.jp (K.M.); antonin.trimaille@laposte.net (A.T.); corina.mirea@chru-strasbourg.fr (C.M.); mariloupeillex@gmail.com (M.P.); sauer.francois@gmail.com (F.S.); cecile.howchoong@chru-strasbourg.fr (C.H.-C.); floriane.zeyons@chru-strasbourg.fr (F.Z.); annie.trinh@chru-strasbourg.fr (A.T.); Helene.petit-eisenmann@chru-strasbourg.fr (H.P.-E.); laurence.jesel-morel@chru-strasbourg.fr (L.J.); patrick.ohlmann@chru-strasbourg.fr (P.O.); 2INSERM (French National Institute of Health and Medical Research), UMR 1260, Regenerative Nanomedicine, FMTS, 67000 Strasbourg, France; 3Neurology Department, Hautepierre Hospital, Strasbourg University Hospital, 67000 Strasbourg, France; olivier.rouyer@chru-strasbourg.fr (O.R.); valerie.wolff@chru-strasbourg.fr (V.W.)

**Keywords:** stroke, transoesophageal echocardiography, imaging

## Abstract

Background and purpose—current guidelines recommend the use of transesophageal echocardiography (TEE) in relation to cardio-embolic sources of stroke. Methods—by using an hospital-based cohort, we retrospectively analyzed consecutive patients with acute ischemic stroke (AIS), acute hemorrhagic stroke (AHS) and transient ischemic attack (TIA) who were admitted in Strasbourg Stroke Center, France between November 2017 to December 2018. TEE reports were screened for detection of potential cardiac sources of embolism and the subsequent change in medical management. We performed univariate and multivariate analyses to identify predictors of relevant TEE findings. Results-out of the 990 patients admitted with confirmed stroke, 432 patients (42.6%) underwent TEE. Patients with TEE were younger (62.8 ± 14.8 vs. 73.8, *p* < 0.001), presented less comorbidities and lower stroke severity assessed by lower NIHSS (2 IQR (0–4) vs. 3 IQR (0–10), *p* < 0.01) and Modified *Rankin Scale* (1 IQR (0–1) vs. 1 (0–3), *p* < 0.01). A total of 227 examinations (52.5%) demonstrated abnormal findings considered as potential cardiac sources of embolism and 31 examinations (7.1%) were followed by subsequent change in medical management. Age (HR: 0.948, 95% CI 0.923 to 0.974; *p* < 0.001), previous AIS (HR: 3.542, 95% CI 1.290 to 9.722; *p* = 0.01), previous TIA (HR: 7.830, CI 95% 2214 to 27,689; *p* = 0.001) and superficial middle cerebral artery territory infarction (HR: 2.774, CI 95% 1.168–6.589; *p* = 0.021) were strong independent predictors with change in medical management following TEE. Conclusions—additional TEE changed the medical course of stroke patients in 7.1% in a French high-volume stroke unit.

## 1. Introduction

Stroke is the third leading cause of death and the leading cause of serious and long-term disability in industrialized countries. With 87% of all strokes of ischemic origin, the Trial of Org 10172 in Acute Stroke Treatment criteria [1] has subdivided acute ischemic stroke (AIS) into 5 subgroups—(i) thrombosis or embolism associated with large vessel atherosclerosis, (ii) cardio-embolic stroke, (iii) lacunar stroke, (iv) other determined cause and (v) cryptogenic cause. New high-quality imaging techniques have produced major changes in the evidence-based treatment of each subgroup of AIS and latest updates from the American Heart Association/American Stroke Association in 2019 recommended echocardiography as a class IIa recommendation to guide patient and treatment selection [2]. As cardio-embolic stroke accounts for 15–30% of AIS [3], the identification of a potential cardiac source of embolism is of paramount importance and both transthoracic (TTE) and/or transesophageal (TEE) echocardiography are the cornerstones in the evaluation of these patients. 

Recent guidelines from both transatlantic cardiovascular societies (ESC and AHA) have tried to address appropriate use and indications of TEE [4,5]. These recommendations came nonetheless with qualifying statements of use in “potentially related” or “suspected sources of cardiac causes” or in cases where it “will alter management”. Together with discrepancies in routine clinical practice [6], the role of TEE in identifying ischemic stroke etiology remains complex and is a matter of ongoing debate [7]. New scoring systems such as the ADAM-C score [8] have tried to better identify patients who are likely to benefit from TEE in stroke units.

The use of percutaneous left atrial appendage (LAA) closure has witnessed a substantial growth in selected patients with nonvalvular atrial fibrillation. TEE may even be performed in the acute phase of hemorrhagic stroke as an imaging modality for pre-procedural LAA closure planning. Description of current « off-label» use of TEE and discussion of the current state and future vision of TEE are important trends and challenges in stroke.

Therefore, we sought to evaluate the prevalence, determinants, feasibility and incremental diagnostic value of TEE use in a real-world practice and cohort of patients referred to a high-volume French stroke unit (SU).

## 2. Methods

### 2.1. Study Design and Population 

We performed a retrospective, single-center, cross-sectional study. Patients with suspected stroke were identified out of 1567 consecutive and unselected patients admitted in Strasbourg Stroke Unit, Strasbourg University, France from November 2017 to December 2018. All patients were evaluated according to current guidelines by standard testing (neuroimaging, Doppler ultrasonography and cardiac investigations). A retrospective hospital-based registry using systematic computer coding of data was conducted using the key words “acute ischemic stroke” (AIS), transient ischemic attack” (TIA) and “acute hemorrhagic stroke” (AHS). The diagnosis of either AIS, TIA and AIS was made according to current recommendations [1]. Patients were excluded if they did not have a documented ischemic, hemorrhagic or transient stroke. The study protocol was approved. All subjects gave their informed consent for inclusion before they participated in the study. The study was conducted in accordance with the Declaration of Helsinki, and the protocol was approved by the institutional review board of the University (CE-2021-7).

### 2.2. Cardiovascular Diagnostic Work-up

In the Strasbourg Stroke Center, standard cardiovascular diagnostic work-up consists for all patients with stroke of 12-lead-electrocardiogram (ECG), in hospital heart rhythm monitoring (telemetry and/or Holter-ECG), TTE and neurovascular ultrasound evaluation for extra- and intracranial arteries. TEE was performed in all patients with AIS or TIA according to current recommendations [1,4,5] unless—(i) a distinct stroke etiology was recognized (e.g., carotid artery stenosis); (ii) TEE findings were unlikely to change medical management (e.g., known atrial fibrillation (AF), total disability and advanced comorbidities) and/or (ii) if the patient refused TEE examination. A specialized Stroke Team daily supervised the ordering of TEEs for hospitalized patients with acute stroke, leveraging the expertise of cardiologists, neurologists, imaging and echo lab specialists.

### 2.3. Transesophageal Echocardiography (TEE)

All patients underwent TTE directly before TEE. TEE was performed according to current guidelines [9] and using a Philips iE33 echocardiographic module and X7-2t probe (Philips medical systems, Andover, Massachusetts). The TEE probes were covered with lidocaine gel and additional conscious sedation was achieved by intravenous administration of benzodiazepines, if needed. TEE examinations were analyzed according to abnormal findings that could explain the etiology of stroke and findings that would change therapy. Abnormal TEE findings included—spontaneous echo contrast (SEC) and thrombi, either atrial and/or ventricular; an atrial septal aneurysm (ASA) defined by the excursion of the ASA ≥10 mm; complex aortic plaques defined as plaques with ≥4-mm-thickness, ulcerated, or associated to mobile thrombi; patent foramen ovale (PFO), a right-to-left shunt (using injected agitated saline solution) diagnosed when microbubbles were detected in the left atrium within 4 cardiac cycles after right atrial opacification. Shunt grading was defined according to the numbers of bubbles seen in a single still frame in the left atrium. The protocol used for shunt grading incorporated 4 grades—Grade 1: <5 bubbles; Grade 2: 5 to 25 bubbles; Grade 3: >25 bubbles and grade 4: opacification of chamber.

For intracardiac right-to-left shunt detection, spontaneous or provoked (during Valsalva) maneuvers were performed. To investigate the effect of TEE findings in the therapeutic management, change in treatment after TEE included (i) anticoagulation initiation (ii) procedural intervention aimed to close an intracardiac shunt, PFO or LAA (iii) cardiac valve replacement (native or prothesis) and (iv) antibiotic initiation for endocarditis. Of note, antiplatelet therapy, antihypertensive medications, and cholesterol-lowering drugs were considered as standard of care for secondary prevention. When patients had prior indications for anticoagulation (e.g., AF, cardiac sources of embolism already visualized with TTE), it was not considered as a change in clinical management. 

After careful reviewing of patients’ medical electronic records and TEE reports, we recorded all TEE findings which were considered as (i) “abnormal TEE findings” and a potential source of cardiogenic embolism and/or (ii) “TEE with change in medical management”. The findings were not recorded if the cardiac source of embolism was already visible in TTE. 

### 2.4. Statistical Analysis 

Continuous variables were expressed as mean ± standard deviation, and categorical variables as frequencies and percentages. Continuous variables between all groups were compared using ANOVA or Kruskal–Wallis test, as appropriate. Pearson’s Chi-squared test was used to compare categorical variables. Continuous variables were analyzed for normal distribution using Shapiro–Wilk test or graphically. Predictive factors of “abnormal TEE findings” and “TEE with change in medical management” were investigated using logistic regression analysis. All tests were two sided. A *p*-value < 0.05 was considered significant. Calculations were performed using SPSS 17.0 (SPSS Inc., Chicago, IL, USA).

## 3. Results

### 3.1. Patients Characteristics

From November 2017 to December 2018, 990 patients with a confirmed diagnosis of stroke were admitted in Strasbourg Stroke Center (Central Illustration and Figure 1). 

A total of 432 hospitalized patients (42.6%) underwent TEE—365 (84.5%) for the evaluation of a documented acute ischemic stroke (AIS), 59 (13.7%) for transient ischemic attack (TIA) and eight (1.8%) for a documented acute hemorrhagic stroke (AHS) (Table 1) (Figure 2). No TEE-related complications were observed in the whole cohort. Patients with TEE were younger (62.8 ± 14.8 vs. 73.8 ± 12.5, *p* < 0.001), less likely to disclose cardiovascular risk factors and less likely to present an history of supraventricular arrhythmia (26 (6.0%) vs. 146 (26.2%), *p* < 0.001), previous AIS (40 (9.3%) vs. 84 (15.1%), *p* = 0.007), ongoing neoplasia or kidney disease (Table 2).

Regarding initial stroke severity, the TEE group showed lower National Institutes of Health Stroke Scale (NIHSS) scores at admission (2 IQR (0–4) vs. 3 IQR (0–10), *p* < 0.01) lower degree of disability/dependence after *stroke assessed by* The Modified *Rankin Scale* (1 IQR (0–1) vs. 1 (0–3), *p* < 0.01), resulting in *shorter length of stay* (9.9 ± 10.6 vs. 13.9 ± 14.4, *p* < 0.001).

### 3.2. Transesophageal Echocardiography (TEE) with Abnormal Findings and TEE with Subsequent Change in Medical Management

Among the 432 TEE performed, 227 exams (52.5%) demonstrated abnormal findings considered as a potential cardiac source of embolism. All TEE findings are summarized in Table 3. PFO was the most common finding (59%) followed by complex aortic plaques (52%) and ASA (21%). Baseline characteristics, biological parameters and stroke scoring systems and scales did not differ according to the presence of abnormal findings in TEE, apart from peripheral artery disease (PAD) history (Table 4). Similarly, all patients underwent TTE evaluation, transcranial Doppler sonography, transcranial color Doppler sonography and ultrasound examination of the neck vessels before TEE and no difference between the two subsets of patients could be shown (Appendix A). 

Among the 432 TEE performed, 31 examinations (7.1%) demonstrated a potential cardiac source of embolism and subsequent change in medical management (Table 5). A total of 16 patients underwent PFO closure (All with shunt grading 3 or 4), six patients underwent left atrial (LA) appendage closure and anticoagulation was initiated for nine patients (three patients with prosthetic valve thrombosis; three patients with LA/Left ventricular thrombus; two patients with LA appendage thrombus and one patient with aortic complex ulcerated plaques).

### 3.3. Predictors of Abnormal Findings in TEE and Subsequent Change in Medical Management following TEE

By univariate Cox analysis, PAD, history of coronary artery disease (CAD) and LA volume assessed by TTE were significant predictors of abnormal findings in TEE. By univariate Cox analysis, age, previous AIS, previous TIA, AF at baseline and de novo AF, anticoagulant therapy at admission, infarct topography and haemoglobin level were significant predictors of TEE with consecutive change in medical management. In multivariate analysis, age (HR: 0.948, 95% CI 0.923 to 0.974; *p* < 0.001), previous AIS (HR: 3.542, 95% CI 1.290 to 9.722; *p* = 0.01), previous TIA (HR: 7.830, CI 95% 2214 to 27,689; *p* = 0.001) and superficial middle cerebral artery territory infarction (HR: 2.774, CI 95% 1.168–6.589; *p* = 0.021) remained strong independent factors associated with change in medical management following TEE (Table 6).

## 4. Discussion

The current report drawn from a cohort of 990 patients tries to clarify the incremental diagnostic and therapeutic value of TEE in a real-world cohort of stroke patients. The salient results of the present study are as follows—1) TEE was performed in 42.6% of cases and remained strongly associated with the initial diagnostic workup of acute ischemic stroke (84.5% of all examinations performed); 2) TEE indication reaches the field of hemorrhagic stroke at the acute phase; 3) TEE was preferably performed in younger and less-disabled patients; 4) 52.5% of TEE examinations demonstrated abnormal findings but only 7.1% TEEs were followed by subsequent change in medical practice with age, previous TIA and superficial middle cerebral artery territory infarction as sole independent predictors associated with change in medical management.

### 4.1. Frequent Abnormal Findings and Few Change in Medical Management

TEE found abnormal findings in 52.5% of the cases. Like any imaging modality, TEE focuses on (i) the description of structural pathological images which can eventually (ii) lead to a therapeutic change. Analogous with incidentalomas, the presence of an atrial septal aneurysm in our cohort (48 patients; 11,1%) led to no change in treatment as antiplatelet therapy, statins and control of cardiovascular risk factors were already the cornerstone of secondary stroke prevention in this case. Only 7.1% of TEE examinations led to a change in medical management. Except for PFO closure, the impact of TEE in our cohort therefore appears modest and reinforces guidelines caution against routine TEE use. The impact of TEE on further change in medical management has recently been the subject of controversy. Despite 35% of TEEs with abnormal findings, Van Woerkom et al. [10] reported a therapeutic change in only 2.5% of the 515 patients included with AIS or TIA. For Rosol et al. [7], change in medical management only occurred in 4% of patients with TEE. Among 1458 consecutive patients admitted for AIS or TIA, Khariton et al. [11] reported a therapeutic change after TEE in 16.7% of cases.

### 4.2. Transesophageal Echocardiography (TEE) in Hemorrhagic Stroke: Towards an Extended Indication and New Horizon for TEE?

Current guidelines from both transatlantic cardiac societies recommend the use of TEE in investigating cardioembolic sources of AIS and TIA. No indication for TEE in the acute phase of hemorrhagic stroke has been so far mentioned in expert consensus documents. As highlighted in our study, TEE now reaches the scope of hemorrhagic stroke. First sought to only assess potential cardioembolic sources of stroke, TEE is now performed at the acute phase of hemorrhagic stroke as an imaging modality for pre-procedural left atrial appendage (LAA) closure planning. Indeed, with the extension in real life practice of LAA closure device implantation even in high risk patients such as those with prior intracranial hemorrhage [12], pre-procedural TEE was performed to define the morphology and dimensions of LAA, assess the shape and size of the ostium, the width of the future landing zone and the length of the LAA.

### 4.3. Futility, Benefit and Applicability of Transesophageal Echocardiography in the Real-World Practice

In our single-centre study, TEE was preferably performed in younger, lower-risk patient with less disability (assessed by NIHSS and the Modified Rankin Scale). This finding highlights the crucial role of the Stroke Team that includes cardiologists, neurologists, imaging and echo lab specialists in assessing every stroke patient and defining adequate diagnostic workup. This raises important questions about the need to identify and acknowledge the possibility of futility in some patients considered for TEE. In the stroke population, a number of factors in addition to traditional risk stratification need to be considered, including age, multimorbidity, fixed disability, frailty and cognition in order to assess the anticipated benefit of TEE and evoked medical changes. Consideration by a multidisciplinary stroke team with broad areas of expertise is critical for assessing likely benefit from TEE. There is currently no existing evidence in current guidelines or validated scoring systems in assessing the potential futility/benefit balance of TEE in stroke patients and this constitutes a gap in evidence in clinical practice. In line with this view, Van Woerkom et al. found that TEE had a low likelihood of a pathologic finding that resulted in a change in treatment strategy, especially in patients ≥80 years of age [10].

Recently, the ADAM-C score was proposed to help physicians identify patients who may benefit from TEE [8]. Based on clinical (age, diabetes, history of CAD, multi-territory stroke) and transthoracic echocardiographic parameters (aortic stenosis from any degree), this score showed interesting diagnostic performance (at a threshold lower than 3—negative predictive value of 95% (CI 94–97)). Applied to our cohort, we found 33.5% of patients with an “informative” TEE but an ADAM-C score <3 who should have been dismissed from TEE evaluation following this score. For TEE exams with change in medical management, 16 patients (48.5% of the cohort) had an ADAM-C score <3 and would not have benefited from TEE. Based on the regression analysis results performed in our cohort study, we emphasized younger age, history of AIS and/or TIA and superficial middle cerebral artery territory infarction among key criteria related to the change in medical management following TEE and stroke. More studies are necessary to better identity patients who are likely to benefit from TEE in stroke units.

## 5. Study Limitations

Our study has several limitations. First, it is a single-center retrospective study with all inherent limitations due to the design of such study. Second, indications to perform a TEE were carried out by a specialized stroke team. As such, patients with a poor prognosis, at extreme ages or patients who refused the examination are either under-represented or absent in the present study. Third, we report a low rate of infective endocarditis (IE) as patients with IE were generally hospitalized in a dedicated Infectious Diseases Intensive Care Unit (IDICU) in Hopitaux Universitaires de Strasbourg. Fourth, local expertise may have influenced our results as strong recommendations regarding PFO closure are still lacking as well the validation for anticoagulation over antiplatelet therapy in complex aortic plaques. As PFO closure represented the majority of patients with a significant change in medical management, the therapeutic impact of TEE may thus be overestimated. Given the relatively small sample size of this study, secondary evaluations of participants with PFO and change in medical practice due to TEE examination are beyond the scope of the data presented. Finally we have no follow-up data and therefore we cannot appreciate the prognostic impact of TEE (morbidity, mortality, risk of recurrent stroke, etc.).

## 6. Conclusions

Additional TEE changed the medical course of stroke patients in 7.1% in a French high-volume stroke unit. The present study supports the need for ongoing and future research aimed to better identity patients who are likely to benefit from TEE in stroke units.

## Figures and Tables

**Figure 1 jcm-10-00805-f001:**
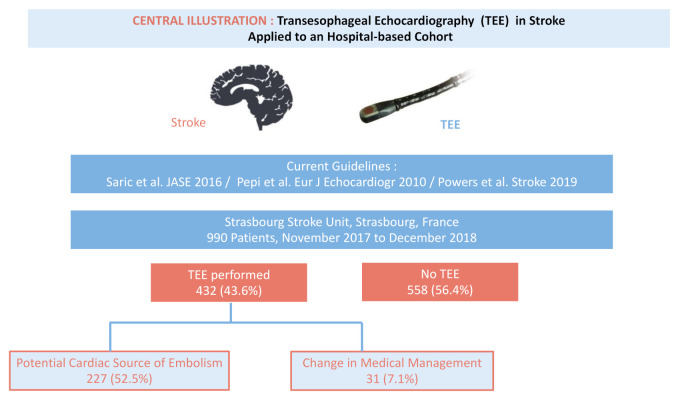
Central illustration. Transesophageal Echocardiography in Stroke Applied to an hospital-based Cohort. Legends—updated European and American guidelines have addressed the use of transesophageal echocardiography (TEE) in the field of stroke. Among 990 patients admitted for stroke in Strasbourg Stroke Center, France, 432 patients (42.6%) underwent TEE. A total of 227 examinations (52.5%) demonstrated abnormal findings and 31 examinations (7.1%) were followed by subsequent change in medical management. Abbreviations—TEE = Transesophageal Echocardiography.

**Figure 2 jcm-10-00805-f002:**
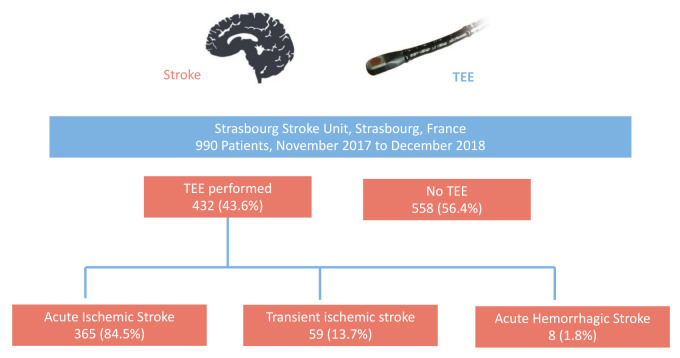
Flow Chart of the Study. Legends—among 990 patients admitted for stroke in Strasbourg Stroke center, France, a total of 432 hospitalized patients (42.6%) underwent TEE—365 (84.5%) for the evaluation of a documented acute ischemic stroke (AIS), 59 (13.7%) for transient ischemic attack (TIA) and 8 (1.8%) for a documented acute hemorrhagic stroke (AHS). Abbreviations—TEE = Transesophageal Echocardiography.

**Table 1 jcm-10-00805-t001:** Transesophageal Echocardiography Examination According to Stroke Etiology.

	Global Population	TEE +	No TEE −	*p*
	(*n* = 990)	(*n* = 432)	(*n* = 558)	
**Types of Stroke**				
Hemorrhagic Stroke—*n* (%)	67 (6.8%)	8 (1.9%)	59 (10.6%)	<0.001
Transient Ischemic Attack—*n* (%)	121 (12.2%)	59 (13.7%)	62 (11.1%)	0.241
Ischemic Stroke—*n* (%)	802(81.0%)	365 (84.5%)	437(78.3%)	0.014
Large-artery atherosclerosis	171	65	106	0.025
Cardioembolism	239	58	181	<0.001
Small-vessel occlusion	7	3	4	1
Stroke of other determined etiology	7	0	7	0.018
Stroke of undetermined etiology	378	239	139	<0.001
**Vascular territory of ischemic stroke**				
Deep middle cerebral artery—*n* (%)	91 (8.2%)	31 (7.2%)	50 (9.0%)	0.350
Superficial middle cerebral artery—*n* (%)	354 (35.8%)	130 (31.9%)	216 (38.7%)	0.032
Borderzone territory—*n* (%)	30 (3.0%)	6 (1.4%)	24 (4.3%)	0.008
Complete middle cerebral artery—*n* (%)	51 (5.2%)	17 (3.9%)	34 (6.1%)	0.148
Anterior cerebral artery—*n* (%)	24 (2.4%)	9 (2.1%)	15 (2.7%)	0.678
Anterior choroidal artery—*n* (%)	61 (6.2%)	23 (5.3%)	38 (6.8%)	0.354
Posterior cerebral artery—*n* (%)	95 (9.6%)	42 (9.7%)	53 (9.5%)	0.914
Vertebro-basilar—*n* (%)	86 (8.7%)	48 (11.1%)	38 (6.8%)	0.022
Cerebellar—*n* (%)	42 (4.2%)	29 (6.7%)	13 (2.3%)	0.001
Multiple—*n* (%)	82 (8.3%)	32 (7.4%)	50(8.9%)	0.060

Data are presented as mean ± or *n* (%).TEE = Transesophageal Echocardiography.

**Table 2 jcm-10-00805-t002:** Patients Characteristics according to Transesophageal Echocardiography Examination.

	Global Population	TEE +	No TEE -	*p*
	(*n* = 990)	(*n* = 432)	(*n* = 558)	
Age (years)	69.03 ± 14.6	62.88 ± 14.8	73.8 ± 12.5	<0.001
Sex (Male)—*n* (%)	553 (55.9%)	263 (60.9%)	290 (52.0%)	0.006
BMI (Kg.m^2^)	26.7 ± 5.4	26.9 ± 5.7	26.4 ± 5.1	0.149
CV Risk Factors				
Hyperte*n*sion—*n* (%)	631 (63.7%)	237 (54.9%)	394 (70.6)	<0.001
Dyslipidemia—*n* (%)	254 (25.7%)	103 (23.8%)	151 (27.1%)	0.271
Past or current smoker—*n* (%)	365 (36.9%)	178 (41.2%)	187 (33.5%)	0.014
Diabetes mellitus—*n* (%)	219 (22.1%)	78 (18.1%)	141 (25.3%)	0.007
Familial history of CAD—*n* (%)	5 (0.5%)	5 (1.2%)	0 (0.0%)	0.016
**Medical History**				
Hemorrhagic stroke—*n* (%)	4 (0.4%)	1 (0.2%)	3 (0.5%)	0.636
Ischemic stroke—*n* (%)	124 (12.5%)	40 (9.3%)	84 (15.1%)	0.007
TIA—*n* (%)	41 (4.1%)	20 (4.6%)	21 (3.8%)	0.523
PAD—*n* (%)	67 (6.8%)	20 (4.6%)	47 (8.4%)	0.021
Carotid endarterectomy—*n* (%)	11 (1.1%)	2 (0.5%)	9 (1.6%)	0.126
AF—*n* (%)	172 (17.4%)	26 (6.0%)	146 (26.2%)	<0.001
*Mechanical valve* *—n (%)*	17 (1.7%)	11 (2.5%)	6 (1.1%)	0.088
Bioprosthetic valve—*n* (%)	4 (0.4%)	1 (0.2%)	3 (0.5%)	0.165
CAD—*n* (%)	118 (11.9%)	50 (11.6%)	68 (12.2%)	0.843
Coronary *Angioplasty—**n* (%)	78 (7.9%)	36 (8.3%)	42 (7.5%)	0.637
STEMI—*n* (%)	47 (4.7%)	22 (5.1%)	25 (4.5%)	0.655
CABG—*n* (%)	12 (1.2%)	5 (1.2%)	7 (1.3%)	1.000
CKD eGFR <60—*n* (%)	68 (6.9%)	21 (4.9%)	47 (8.4%)	0.031
Neoplasia—*n* (%)	121 (12.2%)	40 (9.3%)	81 (14.5%)	0.014
Thromboembolic disease- *n* (%)	55 (5.6%)	11 (2.5%)	44 (7.9%)	<0.001
**Baseline biological parameters**				
Cr eGFR (mL/min/1.73m²) ± DS	77.6 ± 21.7	81.2± 21.6	74.8 ± 21.4	<0.001
Cr level (µmol.L) ± DS	84.7 ± 39.9	88.6 ± 37.4	82.8 ± 40.9	0.058
Total cholesterol (g/L) ± DS	1.9 ± 0.36	1.7 ± 0.45	2.08 ± 0.38	0.447
LDL-C (g/L) ± DS	1.06 ± 0.37	1.10 ± 0.36	1.03 ± 0.38	0.003
HDL-C (g/L) ± DS	0.50 ± 0.32	0.48 ± 0.15	0.50 ± 0.41	0.232
TG (g/L) ± DS	1.23 ± 0.88	1.22 ± 0.68	1.24 ± 1.02	0.759
Glycated Hb(%) ± DS	5.9 ± 1.29	5.9 ± 1.22	5.9 ± 1.34	0.139
**Scoring systems and scales**				
*CHA2DS2* *-*VASc score ***	5.3 ± 1.2	5.1 ± 1.5	5.3 ± 1.2	0.359
HAS BLED Score *	3.4± 1.1	3.3 ± 1.3	3.5± 1.08	0.291
NIHSS (IQR)	2 (0–10)	2(0–4)	3(0–10)	<0.01
Modified *RANKIN* Scale (IQR)	1 (0–3)	0 (0–1)	1 (0–3)	<0.01
Average Lengh of Stay (days)	12.2 ± 13.0	9.9 ± 10.6	13.9 ±14.4	<0.001
**Baseline/** **Pre hospital medication**					
*Antiplatelet* Agents—*n*(%)	309 (31.2%)	119 (27.5%)	190 (34.1%)	0.032
ASA—*n*(%)	275 (27.8%)	110 (25.5%)	165 (29.6)	0.174
Clopidogrel—*n*(%)	83 (8.4%)	33 (7.6%)	50 (9.0%)	0.263
DAPT—*n*(%)	34 (3.4%)	14 (3.2%)	20 (3.6%)	0.456
VKA—*n*(%)	93 (9.4%)	18 (4.2%)	75 (13.4%)	<0.001
DOAC—*n*(%)	59 (6.0%)	13 (3.0%)	46 (8.3%)	<0.001
Beta blockers—*n*(%)	338 (34.1%)	125 (28.9%)	213 (38.2%)	0.001
ACE inhibitors/ARBs—*n* (%)	412 (41.6%)	155 (35.9%)	257 (46.1%)	0.001
Aldosterone-receptor antagonists—*n* (%)	40 (4.0%)	15 (3.5%)	25 (4.5%)	0.264
Statin—*n* (%)	320 (32.3%)	139 (32.2%)	181 (32.4%)	0.493
Furosemide—*n* (%)	93 (9.4%)	21 (4.9%)	72 (12.9%)	<0.001

Data are presented as mean ± or *n* (%). ACE inhibitor = Angiotensin-converting enzyme inhibitor; AF = atrial fibrillation; ARBs = Angiotensin II receptor blockers; ASA = Aspirin; BMI = Body Mass Index; CABG = Coronary Artery Bypass Graft Surgery; CAD = Coronary Artery Disease; CKD = Chronic Kidney Disease; Cr = Creatinine; CV Cardiovascular; DOAC = Direct oral anticoagulant; GFR = glomerular filtration rate; Hb = hemoglobin HDL-C = high-density lipoprotein cholesterol; LDL-C = low-density lipoprotein cholesterol; *NIHSS*
*= *NIH Stroke Scale*/*Score**; PAD = Peripheral Artery Disease; sCr = serum creatinine; STEMI = ST Segment Elevation Myocardial Infarction; TEE = Transesophageal Echocardiography; TIA = Transient Ischemic Attack; thromboembolic disease include venous thromboembolism and/or pulmonary thromboembolism; VKA = Vitamin k antagonists. *** To determine the severity of past and de novo Atrial Fibrillation, the CHA2DS2-VASc score was evaluated in all patients with nonvalvular atrial fibrillation.

**Table 3 jcm-10-00805-t003:** Transesophageal Echocardiography Findings Considered as Potential Cardiogenic Source of Embolism.

	Population with TEE Performed	Abnormal TEE Findings
	(*n* = 432)	(*n* = 227)
Infective Endocarditis—*n* (%)	0 (0.0%)	0 (0.0%)
LA Spontaneous *Contrast*—*n* (%)	14 (3.2%)	14 (6.2%)
LAA Spontaneous *Contrast*—*n* (%)	8 (1.9%)	8 (3.5%)
LA/LV Thrombus—*n* (%)	3 (0.7%)	3 (1.3%)
Left Atrial Appendage Thrombus—*n* (%)	6 (1.4%)	6 (2.6%)
Prosthetic Thrombus—*n* (%)	4 (0.9%)	4 (1.8%)
Patent Foramen Ovale—*n* (%) *	134 (31.0%)	134 (59.0%)
Grade 1: < 5 bubbles	82 (61.2%)	
Grade 2: 5 to 25 bubbles	29 (21.6%)	
Grade 3: > 25 bubbles	13 (9.7%)	
Grade 4: opacification of chamber	10 (7.5)	
Atrial Septal Aneurysm—*n* (%)	48 (11.1%)	48 (21.1%)
Complex aortic plaque—*n* (%)	118 (27.3%)	118 (52.0%)

Data are presented as mean ± or *n* (%). LA = Left atrium; LAA = Left atrial appendage; LV = Left ventricle; TEE = transesophageal echocardiography. * PFO Shunt grading was defined according to the numbers of bubbles seen in a single still frame in the left atrium. The protocol used for shunt grading incorporated 4 grades—Grade 1: <5 bubbles; Grade 2: 5 to 25 bubbles; Grade 3: >25 bubbles and grade 4: opacification of chamber.

**Table 4 jcm-10-00805-t004:** Patients Characteristics according to Considered Potential Cardiac Source of Embolism in Transesophageal Echocardiography.

	TEE	Abnormal TEE Findings	Normal TEE	*p*
	(*n* = 432)	(*n* = 227)	(*n* = 205)	
Age (years)	62.9 ± 14.8	63.0 ±15.3	62.7 ± 14.2	0.859
Sex (Male)—*n* (%)	263 (60.9%)	140 (61.7%)	123 (60.0%)	0.767
BMI (Kg/m^2^)	26.9 ± 5.6	26.9 ± 6.1	26.9 ± 5.1	0.924
**CV Risk Factors**				
Hypertension—*n* (%)	237 (55.0%)	130 (57.3%)	107 (52.2%)	0.333
Dyslipidemia—*n* (%)	103 (23.8%)	56 (24.7%)	47 (22.9%)	0.735
Past or current smoker—*n* (%)	178 (41.2%)	95(41.9%)	83 (40.5%)	0.845
Diabetes mellitus—*n* (%)	78 (18.1%)	45 (19.8%)	33 (16.1%)	0.320
Familial history of CAD—*n* (%)	5 (1.2%)	5(2.2%)	0 (0.0%)	0.063
**Medical History**				
Hemorrhagic stroke—*n* (%)	1 (0.2%)	1 (0.4%)	0 (0.0%)	1.000
Ischemic stroke—*n* (%)	40 (9.3%)	18 (7.9%)	22 (10.7%)	0.324
TIA—*n* (%)	20 (4.6%)	14 (6.2%)	6 (2.9%)	0.168
PAD—*n* (%)	20(4.6%)	17 (7.5%)	3 (1.5%)	0.003
AF—*n* (%)	26 (6.0%)	13 (5.7%)	13 (6.3%)	0.841
*Mechanical valve*—*n* (%)	3 (0.7%)	1 (0.4%)	2 (1.0%)	0.606
Bioprosthetic valve—*n* (%)	1 (0.2%)	0 (0.0%)	1 (0.5%)	0.475
CAD—*n* (%)	50 (11.6%)	29 (12.8%)	21 (10.2%)	0.453
CKD eGFR <60—*n* (%)	21 (4.9%)	9 (4.0%)	12 (5.9%)	0.380
Neoplasia—*n*(%)	40 (9.3%)	21 (9.3%)	19 (9.3%)	1.000
Thromboembolic disease—*n* (%)	11(2.5%)	4 (1.8%)	7 (3.4%)	0.363
**Baseline biological parameters**				
Cr eGFR (mL/min/1.73m²) ± DS	81.2 ± 21.6	80.8± 21.6	81.7 ± 21.5	0.694
Cr level (µmol.L) ± DS	88.6± 37.4	89.8 ± 39.7	87.3± 35.1	0.583
LDL-C (g/L) ± DS	1.10 ± 0.36	1.11 ± 0.39	1.09 ± 0.33	0.504
TG (g/l) ± DS	1.22 ± 0.68	1.23 ± 0.70	1.21 ± 0.67	0.688
Glycated Hb (%) ± DS	5.9 ± 1.22	5.9 ± 1.15	5.9 ± 1.29	0.876
**Scoring systems and scales**				
*CHA2DS2* *-*VASc score**	5.1 ± 1.5	5.3± 1.7	5.0 ± 1.4	0.608
HAS BLED Score	3.3± 1.3	3.6 ± 1.7	3.0± 0.9	0.167
NIHSS (IQR)	2 (0–4)	2 (0–4)	1.5 (0–4)	0.854
Modified *RANKIN* Scale (IQR)	1 (0–1)	0 (0–1)	1 (0–1)	0.441
**Baseline/Pre hospital medication**				
*Antiplatelet* Agents—*n*(%)	119 (27.5%)	66 (29.1%)	53 (25.9%)	0.518
VKA—*n*(%)	18(4.2%)	10 (4.4%)	8 (3.9%)	0.815
DOAC—*n*(%)	125 (28.9%)	69 (30.4%)	56 (27.3%)	0.524

Data are presented as mean ± or *n* (%). AF = atrial fibrillation; BMI = Body Mass Index; CAD = Coronary Artery Disease; CKD = Chronic Kidney Disease; Cr = Creatinine; CV Cardiovascular; DOAC = Direct oral anticoagulant; GFR = glomerular filtration rate; Hb = hemoglobin; LDL-C = low-density lipoprotein cholesterol; *NIHSS*
*=*
*NIH Stroke Scale**/**Score*; PAD = Peripheral Artery Disease; sCr = serum creatinine; TEE = Transesophageal Echocardiography; TIA = Transient Ischemic Attack; thromboembolic disease include venous thromboembolism and/or pulmonary thromboembolism; VKA = Vitamin k antagonists.

**Table 5 jcm-10-00805-t005:** Transesophageal Echocardiography Findings with Treatment Change.

	TEE with Change in Medical Management
	(*n* = 31)
**TEE Findings**	
LA/LV Thrombus—*n*(%)	3 (9.6%)
Left Atrial Appendage Thrombus—*n*(%)	2 (6.4%)
Prosthetic Thrombus—*n*(%)	3 (9.6%)
Left Atrial Appendage Closure—*n*(%)	6 (19.4%)
PFO Closure—*n*(%)	16 (51.6%)
*Ulcerated complex aortic plaque with anticoagulation* *—n(%)*	1 (3.2%)

Data are presented as mean ± or *n* (%). PFO = Patent foramen ovale; TEE = transesophageal echocardiography.

**Table 6 jcm-10-00805-t006:** Predictors of Abnormal Findings in Transesophageal Echocardiography (TEE) And Subsequent Change in Medical Management following TEE.

		Univariate			Multivariate	
	HR	95%CI	*p*	HR	95%CI	*p*
**Abnormal Findings in TEE**						
PAD	5.424	1.566–18.790	0.008	3.144	0.812–12.164	0.097
CAD	3.222	1.167–8.898	0.024	1.921	0.611–6.040	0.264
LA volume (mL/m^2^)	1.014	1.000–1.029	0.045	1.013	0.999–1.027	0.070
**Change In Medical Management following TEE**						
Age (years)	0.967	0.946–0.989	0.003	0.948	0.923–0.974	<0.001
Past ischemic stroke	3.297	1.322–8.224	0.011	3.542	1.290–9.722	0.014
Past TIA	5.013	1.690–14.868	0.004	7.830	2.214–27.689	0.001
AF History	4.400	1.629–11.882	0.003	3.981	0.738–21.468	0.108
*Mechanical valve*	4.243	1.104–16.300	0.035	1.424	0.095–21.299	0.798
VKA at admission	3. 390	1.066–10.781	0.039	1.366	0.115–16.261	0.805
Superficial middle cerebral artery stroke territory	2.083	0.999–4.345	0.050	2.774	1.168–6.589	0.021
Cerebellar stroke territory	3.061	1.080–8.679	0.035	2.687	0.727–9.924	0.138
Baseline Hb (g/dL)	0.827	0.685–0.998	0.047	0.906	0.729–1.126	0.373
De novo AF	3.656	1.456–9.176	0.006	2.582	0.601–11.100	0.202

Data are presented as mean ± or *n* (%). AF = atrial fibrillation; CAD = Coronary Artery Disease; HB = Haemoglobin LA = left atrial; PAD = Peripheral Artery Disease; TEE = transesophageal echocardiography; TIA = Transient Ischemic Attack; VKA = Vitamin k antagonists.

## Data Availability

Data available on request from the authors.

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
