# Peer review of "The Effect of Transoesophageal Echocardiography on Treatment Change in a High-Volume Stroke Unit"

_jcm, 2021, doi:10.3390/jcm10040805_

Round 1
Reviewer 1 Report
Potential benefits from TEE in high volume stroke unit were retrospectively investigated and authors demonstrated their results very well. I think it would be a good study for physicians who are skeptical about wide application of TEE in patients with stroke. If some revisions are made, you might give them a clearer direction. I would like to ask several revisions as below:
- In general, when we are determining whether a given ischemic stroke patient has thromboembolic source, we are trying to seek evidences for subclinical atrial fibrillation in ECG. Could you show their detailed ECG findings?
- In table 1, amongst ischemic stroke patients, can thromboembolic stroke be classified?
- In table 2, the reason for application of CHA2DS2-VASc score, HASBLED score would be explained properly.
- Is table 3 essential for demonstrating results?
- In table 6. P values for Mechanical valve and de novo AF should be checked again. (P = 4.243, P = 3.656, respectively - they might be typos.)
- Grading is important in determining PFO closure. What was the grade of PFOs found/closed in study subjects?
- Rather than just simple comparison between your data and ADAM-C score, it might be more useful to readers to summarise and present the characteristics of patients who should be undergone TEE, based on your regression analysis results.
- It would be more persuasive if authors could show whether the recurrence of stroke could be reduced by treatment changes made after TEE. Could you show data regarding events during follow-up?
Author Response
EDITORIAL BOARD
Comments for Authors
Because your research involved human/animal subjects, please provide us the ethical approval information and approval code in the manuscript.
Ethical Statement is now clearly stated in the text
Changes
All subjects gave their informed consent for inclusion before they participated in the study. The study was conducted in accordance with the Declaration of Helsinki, and the protocol was approved by the institutional review board of the University (CE-2021-7).
REVIEWER 1
Comments and Suggestions for Authors
Potential benefits from TEE in high volume stroke unit were retrospectively investigated and authors demonstrated their results very well. I think it would be a good study for physicians who are skeptical about wide application of TEE in patients with stroke. If some revisions are made, you might give them a clearer direction. I would like to ask several revisions as below:
- In general, when we are determining whether a given ischemic stroke patient has thromboembolic source, we are trying to seek evidence for subclinical atrial fibrillation in ECG. Could you show their detailed ECG findings?
We thank the reviewer for evaluating our manuscript and we appreciate her/his valuable comments. As requested, patient's ECG characteristics on admission (Sinus rhythm or AF) and new AF diagnosed in the stroke unit using electrocardiographic monitoring for >24 hours have given below and added to the manuscript. In line with these findings, we have to acknowledge the following limitation: providing longer duration of monitoring and results is beyond the scope of this study. Our retrospective, single-center, cross-sectional study was derived from on an hospital-based cohort and longer duration of monitoring (e.g > 3 months, REVEAL dataset etc.) that is associated with an increased detection of AF and cardioembolic source (AHA and ESC guidelines; Dussault et al. Review. Circ Arrhythm Electrophysiol) is not given consideration in our manuscript.
Changes
|
TEE (n = 432) |
Abnormal TEE findings (n =227) |
Normal TEE (n = 205) |
p |
|
|
ECG at admission |
||||
|
AF |
26 (6%) |
15 (6,6%) |
11 (5,4%) |
0,588 |
|
Sinus Rythm |
406 (94%) |
212 (93,4%) |
194 (94.6%) |
0,588 |
|
In hospital settings |
||||
|
New AF |
16 (3,7%) |
7 (3,1%) |
9 (4,4%) |
0,473 |
2.In table 1, amongst ischemic stroke patients, can thromboembolic stroke be classified?
Following the reviewer request and using the TOAST classification, thromboembolic stroke was classified according to five subtypes
Large-artery atherosclerosis
Cardioembolism
Small-vessel occlusion
Stroke of other determined etiology
Stroke of undetermined etiology
Changes
Table 1. Transesophageal Echocardiography Examination According to Stroke Etiology.
|
Global population |
TEE + |
No TEE - |
p |
||||
|
(n=990) |
(n=432) |
(n=558) |
|||||
|
|
|
||||||
|
Types of Stroke |
|||||||
|
Hemorrhagic Stroke -n(%) |
67 (6.8%) |
8 (1.9%) |
59 (10.6%) |
<0.001 |
|||
|
Transient Ischemic Attack -n(%) |
121 (12.2%) |
59 (13.7%) |
62 (11.1%) |
0.241 |
|||
|
Ischemic Stroke -n(%) |
802(81.0%) |
365 (84.5%) |
437(78.3%) |
0.014 |
|||
|
Large-artery atherosclerosis 171 65 106 0.025 Cardioembolism 239 58 181 <0.001 Small-vessel occlusion 7 3 4 1 Stroke of other determined etiology 7 0 7 0.018 Stroke of undetermined etiology 378 239 139 <0.001 |
|||||||
|
|
|||||||
3.In table 2, the reason for application of CHA2DS2-VASc score, HASBLED score would be explained properly.
To determine the severity of past and de novo AF, the CHA2DS2-VASc score was evaluated in all patients with nonvalvular AF. We thank you for this clear point, and we have now addressed this data in table 2
Table 2. Patients Characteristics according to Transesophageal Echocardiography Examination.
|
Global Population |
TEE + |
No TEE - |
p |
|
|||
|
(n=990) |
(n=432) |
(n=558) |
|
||||
|
|
|
||||||
|
Scoring systems and scales |
|
||||||
|
CHA2DS2-VASc score *
|
5.3±1.2 |
5.1±1.5 |
5.3±1.2 |
0.359 |
|
||
|
HAS BLED Score * |
3.4± 1.1 |
3.3±1.3 |
3.5± 1.08 |
0.291 |
|
||
|
Average Lengh of Stay (days) |
12.2±13.0 |
9.9±10.6 |
13.9 ±14.4 |
<0.001 |
|
||
|
Baseline/ Pre hospital medication |
|||||||
Data are presented as mean ± or n (%). ACE inhibitor = (………..) VKA = Vitamin k antagonists.
* To determine the severity of past and de novo atrial fibrillation, the CHA2DS2-VASc score was evaluated in all patients with nonvalvular atrial fibrillation.
4.Is table 3 essential for demonstrating results?
The purpose of our study was to determine the incidence of potential sources of cerebro-embolism, and the extent to which the TEE findings altered management. We demonstrated a high incidence of potential sources of cerebro-embolism (≈ 50%) as highlighted in table 3. Primarily PFO and complex aortic atheroma, but we found that the findings infrequently changed clinical management (7.1%).
We believe that table 3 has the important merit of providing a good descriptive analysis of the findings which can be expected to be found on TEE performed for evaluation of stroke or TIA in a contemporary clinical population selected by fairly typical clinical criteria (lack of other obvious etiology for stroke). Table 3 highlights the disparate nature of the TEE findings and may even raise questions/dilemma for future research to be conducted. Indeed, abnormal TEE findings include spontaneous echo contrast which is a very common finding but poorly correlates with stroke risk (as opposed to "sludge" or an atrial thrombus). Similarly, we formally disclose in table 3 no infective endocarditis (EI) as such patients with IE were hospitalized in a dedicated Infectious Diseases Intensive Care Unit (IDICU).
Altogether, table 3 provides a compressive, rapid and synthetic description of detailed findings which can be expected to found on TEE performed for evaluation of stroke or TIA.
Table 3. Transesophageal Echocardiography Findings Considered as Potential Cardiogenic Source of Embolism.
|
|
Population with TEE performed |
Abnormal TEE findings |
|||
|
(n=432) |
(n=227) |
||||
|
|
|
|
|||
|
Infective Endocarditis -n(%) |
0 (0.0%) |
0 (0.0%) |
|||
|
LA Spontaneous Contrast -n(%) |
14 (3.2%) |
14 (6.2%) |
|||
|
LAA Spontaneous Contrast -n(%) |
8 (1.9%) |
8 (3.5%) |
|||
|
LA/LV Thrombus -n(%) |
3 (0.7%) |
3 (1.3%) |
|||
|
Left Atrial Appendage Thrombus -n(%) |
6 (1.4%) |
6 (2.6%) |
|||
|
Prosthetic Thrombus -n(%) |
4 (0.9%) |
4 (1.8%) |
|||
|
Patent Foramen Ovale -n(%) |
134 (31.0%) |
134 (59.0%) |
|||
|
Atrial Septal Aneurysm -n(%) |
48 (11.1%) |
48 (21.1%) |
|||
|
Complex aortic plaque -n(%) |
118 (27.3%) |
118 (52.0%) |
|
||
Data are presented as mean ± or n (%). LA = Left atrium; LAA = Left atrial appendage; LV = Left ventricle; TEE = transesophageal echocardiography.
5.In table 6. P values for Mechanical valve and de novo AF should be checked again. (P = 4.243, P = 3.656, respectively – they might be typos.)
We thank the reviewer for bringing this critical point to our attention. The typo error has been corrected.
Table 6. Predictors Of Abnormal Findings in Transesophageal Echocardiography (TEE) And Subsequent Change In Medical Management following TEE.
|
|
|
Univariate |
|
|
|
Multivariate |
|
|||
|
|
HR |
95%CI |
p |
HR |
95%CI |
p |
|
|||
|
Abnormal Findings in TEE |
|
|
|
|
|
|
|
|||
|
PAD |
5.424 |
1.566-18.790 |
0.008 |
3.144 |
0.812-12.164 |
0.097 |
|
|||
|
CAD |
3.222 |
1.167-8.898 |
0.024 |
1.921 |
0.611-6.040 |
0.264 |
|
|||
|
LA volume (ml/m2) |
1.014 |
1.000-1.029 |
0.045 |
1.013 |
0.999-1.027 |
0.070 |
|
|||
|
Change In Medical Management following TEE
|
|
|
|
|
|
|||||
|
Age (years) |
0.967 |
0.946-0.989 |
0.003 |
0.948 |
0.923-0.974 |
<0.001 |
|
|||
|
Past ischemic stroke |
3.297 |
1.322-8.224 |
0.011 |
3.542 |
1.290-9.722 |
0.014 |
|
|||
|
Past TIA |
5.013 |
1.690-14.868 |
0.004 |
7.830 |
2.214-27.689 |
0.001 |
|
|||
|
AF History |
4.400 |
1.629-11.882 |
0.003 |
3.981 |
0.738-21.468 |
0.108 |
|
|||
|
Mechanical valve |
4.243 |
1.104-16.300 |
4.2430.035 |
1.424 |
0.095-21.299 |
0.798 |
|
|||
|
VKA at admission |
3. 390 |
1.066-10.781 |
0.039 |
1.366 |
0.115-16.261 |
0.805 |
|
|||
|
Superficial middle cerebral artery stroke territory |
2.083 |
0.999-4.345 |
0.050 |
2.774 |
1.168-6.589 |
0.021 |
|
|||
|
Cerebellar stroke territory
|
3.061 |
1.080-8.679 |
0.035 |
2.687 |
0.727-9.924 |
0.138 |
|
|||
|
Baseline Hb (g/dl) |
0.827 |
0.685-0.998 |
0.047 |
0.906 |
0.729-1.126 |
0.373 |
|
|||
|
De novo AF |
3.656 |
1.456-9.176 |
3.656 0.006 |
2.582 |
0.601-11.100 |
0.202 |
|
|||
|
|
|
|
|
|
|
|
|
|||
Data are presented as mean ± or n (%). AF = atrial fibrillation; CAD = Coronary Artery Disease; HB = Haemoglobin LA = left atrial; PAD = Peripheral Artery Disease; TEE = transesophageal echocardiography; TIA = Transient Ischemic Attack; VKA = Vitamin k antagonists.
- Grading is important in determining PFO closure. What was the grade of PFOs found/closed in study subjects?
We thank the reviewer for bringing this critical point to our attention.
To date, there is no single widely accepted grading scheme for assessing the degree of left-to-right shunt from a PFO according to current guidelines (ASE Guidelines & Standards; Silvestry FE et al. Guidelines for the echocardiographic assessment of atrial septal defect and patent foramen ovale: from the American Society of Echocardiography and Society for cardiac angiography and interventions). Furthermore, this report states that « attempted quantification of right-to-left shunting based on the number of microbubbles appearing in the left heart on an echocardiographic still frame; however, this number is dependent on the amount of microbubbles injected and the adequacy of the Valsalva maneuver ».
Mimicking the proposed grading scheme, published by Rana et al in a State-of-the-Art Paper published in JACC Cardiovascular Imaging, we sought to better depict PFO grading, according to the reviewer’s request. Indeed, shunt grading was defined according to the numbers of bubbles seen in a single still frame in the left atrium. The protocol used for shunt grading incorporates 4 grades:
- Grade 1: <5 bubbles
- Grade 2: 5 to 25 bubbles
- Grade 3: >25 bubbles
- And grade 4: opacification of chamber
Changes
Addition of two sentences
- Methods section: Transesophageal Echocardiography (TEE)
Shunt grading was defined according to the numbers of bubbles seen in a single still frame in the left atrium. The protocol used for shunt grading incorporated 4 grades: Grade 1: <5 bubbles; Grade 2: 5 to 25 bubbles; Grade 3: >25 bubbles and grade 4: opacification of chamber
- Results section table 3 + One sentence
Table 3. Transesophageal Echocardiography Findings Considered as Potential Cardiogenic Source of Embolism.
|
|
Population with TEE performed |
Abnormal TEE findings |
|||
|
(n=432) |
(n=227) |
||||
|
|
|
|
|||
|
Infective Endocarditis -n(%) |
0 (0.0%) |
0 (0.0%) |
|||
|
LA Spontaneous Contrast -n(%) |
14 (3.2%) |
14 (6.2%) |
|||
|
LAA Spontaneous Contrast -n(%) |
8 (1.9%) |
8 (3.5%) |
|||
|
LA/LV Thrombus -n(%) |
3 (0.7%) |
3 (1.3%) |
|||
|
Left Atrial Appendage Thrombus -n(%) |
6 (1.4%) |
6 (2.6%) |
|||
|
Prosthetic Thrombus -n(%) |
4 (0.9%) |
4 (1.8%) |
|||
|
Patent Foramen Ovale -n(%)* |
134 (31.0%) |
134 (59.0%) |
|||
|
Grade 1: < 5 bubbles Grade 2: 5 to 25 bubbles Grade 3 : > 25 bubbles Grade 4 : opacification of chamber |
82 (61.2%) 29 (21.6%) 13 (9.7%) 10 (7.5) |
|
|||
|
Atrial Septal Aneurysm -n(%) |
48 (11.1%) |
48 (21.1%) |
|||
|
Complex aortic plaque -n(%) |
118 (27.3%) |
118 (52.0%) |
|
||
Data are presented as mean ± or n (%). LA = Left atrium; LAA = Left atrial appendage; LV = Left ventricle; TEE = transesophageal echocardiography. *PFO Shunt grading was defined according to the numbers of bubbles seen in a single still frame in the left atrium. The protocol used for shunt grading incorporated 4 grades: Grade 1: <5 bubbles; Grade 2: 5 to 25 bubbles; Grade 3: >25 bubbles and grade 4: opacification of chamber
16 patients underwent PFO closure (All with shunt grading 3 or 4), 6 patients underwent left atrial (LA) appendage closure and anticoagulation was initiated for 9 patients (3 patients with prosthetic valve thrombosis; 3 patients with LA/Left ventricular thrombus; 2 patients with LA appendage thrombus and 1 patient with aortic complex ulcerated plaques).
7.Rather than just simple comparison between your data and ADAM-C score, it might be more useful to readers to summarize and present the characteristics of patients who should be undergone TEE, based on your regression analysis results.
We appreciate the valuable comment from the reviewer. This observation is indeed missing from the discussion submitted to her/his evaluation. Change has been made according to the reviewer’s request: on top of the ADAM-C Score comparison; a dedicated rationalization has been added to ease the reading experience of JCM readers.
Changes
Addition of one sentence
Based on the regression analysis results performed in our cohort study, we emphasized younger age, history of AIS and/or TIA and superficial middle cerebral artery territory infarction among key criteria related to the change in medical management following TEE and stroke. More studies are necessary to better identity patients who are likely to benefit from TEE in stroke units.
8.It would be more persuasive if authors could show whether the recurrence of stroke could be reduced by treatment changes made after TEE. Could you show data regarding events during follow-up?
We would like to thank the Reviewer for her/his pertinent comment. Follow-up data would have undoubtedly improved our dataset and analysis. However, follow-up data are unavailable due to the primary design of our study. Such comparison would be of paramount relevance but deserves additional board agreements and analyses including electronic hospital record data and/or patient phone call interviews at a dedicated censoring time.
To acknowledge these limitations, it is stated in the text
“Given the relatively small sample size of this study, secondary evaluations of participants with PFO and change in medical practice due to TEE examination are beyond the scope of the data presented. Finally, we have no follow-up data and therefore we cannot appreciate the prognostic impact of TEE (morbidity, mortality, risk of recurrent stroke etc..)
Reviewer 2 Report
1: This is a decent manuscript overall. I believe the use of TEE in haemorrhagic stroke patients could be better presaged in the introduction- as the authors noted this is not currently standard practice and the paper would read better if the reasons for including haemorrhagic stroke patients were described early.
2: The authors state on line 64
“We performed a retrospective, single-center, cross-sectional study” but shortly threafter
“A prospective hospital-based registry using systematic 69 computer coding of data was conducted using the key words “acute ischemic stroke” (AIS), transient 70 ischemic attack” (TIA) and “acute hemorrhagic stroke” (AHS)”.
So was the study prospective or retrospective?
3: Baseline variables such as the NIHSS and mRS would be better presented as median (interquartile range) rather than mean and standard deviation, and differences between groups for these variables would be better tested for significance using the Mann-Whitney U test.
4: On a formatting note, it would be best if the Tables did not break across pages wherever possible- I'm sure this can be dealt with in final proofing.
Finally on a minor wording note, line 162 appears to be missing a word (most likely "differ" or "vary").
Author Response
EDITORIAL BOARD
Comments for Authors
Because your research involved human/animal subjects, please provide us the ethical approval information and approval code in the manuscript.
Ethical Statement is now clearly stated in the text
Changes
All subjects gave their informed consent for inclusion before they participated in the study. The study was conducted in accordance with the Declaration of Helsinki, and the protocol was approved by the institutional review board of the University (CE-2021-7).
REVIEWER 2
Comments and Suggestions for Authors
1: This is a decent manuscript overall. I believe the use of TEE in haemorrhagic stroke patients could be better presaged in the introduction- as the authors noted this is not currently standard practice and the paper would read better if the reasons for including haemorrhagic stroke patients were described early.
We thank the reviewer for evaluating our manuscript and we appreciate her/his valuable comments. As noticed by the reviewer, the use of TEE in haemorrhagic stroke patients is a “grey zone” beyond the scope of current practice guidelines. Indeed, our analysis benefited from; and was limited at the same time by the inclusion of hemorrhagic stroke patients, which represented a small but clinically distinct population. We definitely believe that haemorrhagic stroke will be an extended indication and a new horizon for TEE, as people get older, with increased prevalence of AF, and dedicated new therapeutic strategies and indications for LAA closure available in order to avoid anticoagulants.
According to the reviewer’s request, the introduction now better reflects such important trends and challenges and it better motivates the rationale behind our analysis.
Changes - Introduction
“The use of percutaneous left atrial appendage (LAA) closure has witnessed a substantial growth in selected patients with nonvalvular atrial fibrillation. TEE may even be performed at the acute phase of hemorrhagic stroke as an imaging modality for pre-procedural LAA closure planning. Description of current « off-label» use of TEE and discussion of the current state and future vision of TEE are important trends and challenges in stroke”
2: The authors state on line 64 “We performed a retrospective, single-center, cross-sectional study” but shortly threafter
“A prospective hospital-based registry using systematic 69 computer coding of data was conducted using the key words “acute ischemic stroke” (AIS), transient 70 ischemic attack” (TIA) and “acute hemorrhagic stroke” (AHS)”. So was the study prospective or retrospective?
Here, we apologize for this serious typing error. This sentence has been rephrased, cautiously.
“A retrospective hospital-based registry using systematic computer coding of data was conducted using the key words “acute ischemic stroke” (AIS), transient ischemic attack” (TIA) and “acute hemorrhagic stroke” (AHS)”
3: Baseline variables such as the NIHSS and mRS would be better presented as median (interquartile range) rather than mean and standard deviation, and differences between groups for these variables would be better tested for significance using the Mann-Whitney U test
According to the reviewer’s request. Results do remain the same after this additional analysis
Changes in the text and dedicated tables
Abstract Section
Patients with TEE were younger (62.8±14.8 vs 73.8, p <0.001), presented less comorbidities and lower stroke severity assessed by lower NIHSS (2 IQR (0-4) vs 3 IQR (0-10), p<0.01) and Modified Rankin Scale (1 IQR (0-1) vs 1 (0-3), p <0.01).
Results Section
Regarding initial stroke severity, the TEE group showed lower National Institutes of Health Stroke Scale (NIHSS) scores at admission (2 IQR (0-4) vs 3 IQR (0-10), p<0.01) lower degree of disability/dependence after stroke assessed by The Modified Rankin Scale (1 IQR (0-1) vs 1 (0-3), p <0.01) resulting in shorter length of stay ((9.9 ± 10.6 vs 13.9 ± 14.4, p<0.001).
4: On a formatting note, it would be best if the Tables did not break across pages wherever possible- I'm sure this can be dealt with in final proofing. Finally, on a minor wording note, line 162 appears to be missing a word (most likely "differ" or "vary").
We thank the reviewer for her/his comment. We fully agree and hope that in case of acceptance, the production process and the journal production team will be able to deal with the reviewer’s editing concern.
Typo error line 162 has been changed according to the reviewer’s request.
Changes
“Baseline characteristics, biological parameters and stroke scoring systems and scales did not differ according to the presence of abnormal findings in TEE, apart from peripheral artery disease (PAD) history (Table 4).”